# Carbon Dot Emission Enhancement in Covalent Complexes with Plasmonic Metal Nanoparticles

**DOI:** 10.3390/nano13020223

**Published:** 2023-01-04

**Authors:** Irina A. Arefina, Danil A. Kurshanov, Anna A. Vedernikova, Denis V. Danilov, Aleksandra V. Koroleva, Evgeniy V. Zhizhin, Aleksandr A. Sergeev, Anatoly V. Fedorov, Elena V. Ushakova, Andrey L. Rogach

**Affiliations:** 1International Research and Education Centre for Physics of Nanostructures, ITMO University, Saint Petersburg 197101, Russia; 2Interdisciplinary Resource Centre for Nanotechnology, Saint Petersburg State University, Saint Petersburg 199034, Russia; 3Centre for Physical Methods of Surface Investigation, Saint Petersburg State University, Saint Petersburg 199034, Russia; 4Department of Physics, Hong Kong University of Science and Technology, Hong Kong SAR 999077, China; 5Department of Materials Science and Engineering, Centre for Functional Photonics (CFP), City University of Hong Kong, Hong Kong SAR 999077, China

**Keywords:** carbon dots, metal nanoparticles, carbodiimide chemistry, plasmonic resonance, emission enhancement

## Abstract

Carbon dots can be used for the fabrication of colloidal multi-purpose complexes for sensing and bio-visualization due to their easy and scalable synthesis, control of their spectral responses over a wide spectral range, and possibility of surface functionalization to meet the application task. Here, we developed a chemical protocol of colloidal complex formation via covalent bonding between carbon dots and plasmonic metal nanoparticles in order to influence and improve their fluorescence. We demonstrate how interactions between carbon dots and metal nanoparticles in the formed complexes, and thus their optical responses, depend on the type of bonds between particles, the architecture of the complexes, and the degree of overlapping of absorption and emission of carbon dots with the plasmon resonance of metals. For the most optimized architecture, emission enhancement reaching up to 5.4- and 4.9-fold for complexes with silver and gold nanoparticles has been achieved, respectively. Our study expands the toolkit of functional materials based on carbon dots for applications in photonics and biomedicine to photonics.

## 1. Introduction

Carbon dots (CDs) are a kind of nano-sized fluorescent carbon materials with attractive optical properties, such as excitation-dependent (or independent) light emission with a high photoluminescence quantum yield (PL QY) [1]. CDs have been shown to be biocompatible [2] and/or low toxic materials [3], with an attractive catalytic performance [4] and high chemical and photostability [5], which can conveniently be produced by a low-cost chemical synthesis [6]. Overall, the properties of CDs are considered promising for a wide range of applications in biomedicine [7], sensing [8], catalysis [9], and photonics [10]. The surface of CDs can be functionalized via electrostatic interactions with other (charged) molecules [11,12,13], or covalently by performing carbodiimide coupling, esterification, sulfonation, or copolymerization. The molecular moieties covalently bonded to the surface of CDs can significantly influence their energy structure and optical properties, eventually resulting in the shift of optical transitions [14] and the emergence of chiral signals [15], etc. In particular, carbodiimide coupling can lead to the formation of stable complexes of CDs with molecules, other nanoparticles, and even drugs, offering new prospects for bioimaging [16], detection [17], and drug delivery [18,19]. Since CDs typically possess rather large Stokes shifts, i.e., they absorb in UV and blue spectral regions and emit in green to NIR spectral regions [20,21], the colloidal CD complexes with silver or gold NPs can be implemented for bio-visualization and sensing. CDs can be coupled with noble metal nanoparticles (MNPs) such as Au, Ag, and Pt, or transition metals such as Cu, Fe, and Zn, resulting in either enhanced emission [22] and/or improved catalytic properties [23]. Herein, we focused on several combinations of the existing colloidal systems of CD with MNP. Generally, quantum emitters located near MNPs may experience several effects: (i) the local enhancement of PL excitation; (ii) the acceleration of spontaneous recombination rate related to the Purcell effect [24]; and (iii) an increase in non-radiative decay due to the plasmon-induced energy transfer [25]. The (i) and (ii) processes enhance emission, while the (iii) process quenches it, as schematically illustrated in Figure 1a. The impact of each of these processes on the overall PL change depends on multiple parameters: the morphology of the MNP; the distance between the emitter and the MNP; the overlap of their spectra (absorption or emission of the fluorophore with plasmon resonance of the MNP), as shown in Figure 1b; and the dielectric constant of the surrounding media. The distance r between the CD and MNP, which can be controlled by the length of the CD–MNP bond (Figure 1c), is responsible for near-field plasmon-induced PL quenching, including surface energy transfer [26], and affects the PL intensity proportional to $1/r^{n}$ [27]. In contrast to the Förster resonant energy transfer (FRET), where *n* = 6, here, the value of n is either 3 or 4 [28]. In the resonant excitation regime (Figure 1b-i), where the excitation wavelength resonantly overlaps with the plasmon resonance of the MNP, the PL enhancement would occur due to the local amplification of electromagnetic field by MNP and the corresponding increase in the probability for CD to absorb light. On the other hand, the overlap of the plasmon resonance with the PL band of CDs can maximize light–matter coupling through the Purcell effect, accelerating the spontaneous recombination rate of the CD, which can compensate or at least reduce the plasmon-induced PL quenching (Figure 1b-ii). Thus, the possible enhancement of the PL QY of CDs, which has great importance for bioimaging and sensing applications [29], could be achieved by the design of the CD–MNP complex, e.g., the interaction of the nanoparticles and the overlap between the absorption/PL bands of CDs with the plasmon resonance of MNPs (Figure 1).

We have studied several CD–MNP interactions in dynamic and covalently bonded complexes (see Figure 1c). For CD–MNP formed via carbodiimide chemistry, depending on the overlap between CD’s absorption and emission with MNP’s plasmon resonance, the emission of CDs was enhanced up to 5.4 and 4.9 times when coupled with silver and gold nanoparticles, which was accompanied by PL decay time decrease, thus indicating the metal-enhanced fluorescence.

## 2. Materials and Methods

### 2.1. Materials

Benzoic acid (≥99.5%), cysteamine hydrochloride (≥98%), gold (III) chloride trihydrate (HAuCl_4_) (≥99.9%), *N*,*N*-dimethylformamide (DMF) (99,80%), *N*-(3-dimethylaminopropyl)-*N*′-ethylcarbodiimide hydrochloride (EDC) (≥98%), *N*-hydroxysuccinimide (NHS) (98%), silver nitrate (AgNO_3_) (≥99.0%), sodium borohydride (NaBH_4_) (99%), o-phenylenediamine (99.5%), and polyethylene glycol (PEG, MW = 300) were purchased from Sigma-Aldrich. Ethanol (>96%) was purchased from Vekton (Penfield, NY, USA). Water was purified using a Milli-Q (18.2 MΩ) reagent-grade water system from Millipore (Burlington, MA, USA).

### 2.2. Sample Preparation

Synthesis of CDs: CD-1 was synthesized by the hydrothermal method as described in Ref. [30]. CD-2 was synthesized by a modified solvothermal method of Ref. [31]. CD-3 was synthesized according to Ref. [32]. The details of the synthesis procedures are given in the Appendix A.

Synthesis of MNPs: The synthesis of Au-PEG NPs was adopted from Ref. [33]. The synthesis of Au-PEG NPs was adopted from Ref. [33]. Silver nanoparticles stabilized with cysteamine were synthesized according to Ref. [34]. Gold nanoparticles stabilized with cysteamine were synthesized according to Ref. [35]. The details of the synthesis procedures are given in the Appendix A.

Formation of the CD–MNP complexes: The conjugation of CDs with Ag or Au NPs was performed by carbodiimide coupling (EDC/NHS reaction). A mixture of 2 mL aqueous solution of EDC (1 mg/mL) and 1 mL solution of the CDs (4 mg/mL) in 2 mL of phosphate-buffered saline (PBS) was stirred for 15 min; after that, a 2 mL solution of NHS (1 mg/mL) was added and the mixture was stirred for another 15 min. Finally, 1 mL of stock solution of Ag or Au NPs was added quickly and the resulting mixture was stirred for 24 h.

### 2.3. Experimental Setup

The morphology of the synthesized nanoparticles and their complexes was studied using a MERLIN TEM (Carl Zeiss, Oberkochen, Germany). XPS measurements were performed on an ESCALAB 250Xi photoelectron spectrometer with AlKα radiation (photon energy of 1486.6 eV). FTIR spectra were recorded on a Tenzor II infrared spectrophotometer (Bruker, Billerica, MA, USA). DLS measurements were carried out on a particle size analyzer (Zetasizer Nano ZS, Malvern Instruments Ltd., Malvern, UK). The absorption and PL spectra were collected on a UV-3600 spectrophotometer (Shimadzu, Kyoto, Japan) and an FP-8200 spectrofluorometer (Jasco, Oklahoma City, OK, USA), respectively. Rhodamine 6G was used as a reference dye to determine PL QYs. PL decays were measured on a MicroTime 100 system (PicoQuant, Berlin, Germany) with an excitation at 405 nm and signal collection in the spectral range from 430 to 780 nm. All PL decay curves were fitted using biexponential function, and the average PL decay time of the samples was calculated as $\tau =\left(A_{1}\cdot \tau_{1}^{2}+A_{2}\cdot \tau_{2}^{2}\right)/\left(A_{1}\cdot \tau_{1}+A_{2}\cdot \tau_{2}\right)$, where *A* and *τ* are the amplitude and PL decay time of each component in biexponential function, respectively. The optical cross-section and field enhancement of the MNPs were calculated via Mie theory according to the procedure described in Ref. [36] using the NFMie MATLAB calculation tool (LP2L, Numerical tools—NFMie program, http://lp2l.polymtl.ca/en/outils (accessed on 8 December 2022)). Extinction, absorption, and scattering cross-sections were calculated for the mean diameter of the particle in water, according to data in Appendix A. For 2D E-Field intensity distribution, the incident light with E_0_ = 1 intensity propagated across the *Z*-axis; the simulation was running in the X-Z plane.

## 3. Results

### 3.1. Dynamic Interaction of CDs and MNPs

We first studied CDs with different optical properties interacting with gold NPs in colloidal solution, where their interaction was governed by the dynamic attraction of their oppositely charged surfaces. Gold NPs stabilized with polyethylene glycol (PEG) were synthesized according to Ref. [33] and designated hereafter as Au-PEG NPs. Two kinds of CDs—CD-1 synthesized according to Ref. [30], with absorption at 345 nm and PL band at 475 nm (Figure 2a), and CD-2 synthesized according to Ref. [31], with absorption at 425 nm and PL band at 535 nm (Figure 2d)—were chosen. The Au-PEG NPs had a broad plasmon peak at 530 nm (Figure 2a,d). For both types of CDs, an increase in PL intensity upon the increase in Au-PEG NP concentration was observed (Figure 2b,e). For the sample CD-1 mixed with Au-PEG NPs, emission quenching was observed upon the addition of up to 20 μL of Au-PEG NPs, while starting from 80 μL, a PL recovery took place, which reached 5% of the initial PL intensity (Figure 2c). The PL lifetimes of CD-1 with and without Au-PEG NPs were very similar at 11.5 and 10.5 ns, respectively. For the sample CD-2 mixed with Au-PEG NPs, the shorter-wavelength part of the PL spectrum at around 450 nm was quenched, while the longer-wavelength part of the PL was enhanced (Figure 2e) up to 10% (Figure 2f). The larger PL enhancement observed for the Au-PEG NP complexes with longer-wavelength absorbing and emitting CD-2 (Figure 2c,f) may be related to the simultaneous effect of the MNP-assisted enhancement in the CDs’ absorption and the PL band overlapping with the plasmon resonance of the Au-PEG NPs. The PL lifetimes of CD-2 with and without Au-PEG NPs were the same (5.8 ns), which can be related to weak light–matter coupling.

It should be noted that the dynamic interaction of the oppositely charged surface of these nanoparticles (CDs and MNPs) may not necessarily result in the formation of stable complexes, but rather lead to undesired aggregation and precipitation. Thus, we proceeded with the formation of covalently bonded CD–MNP complexes using CDs with an even more redshifted PL band. For that, we used CDs designated as CD-3 with a PL maximum at 570 nm and cysteamine-capped silver or gold MNPs (designated as AgNPs and AuNPs). The synthesis procedures of all nanoparticles are provided in the Appendix A.

### 3.2. Covalent Complexes Based on CDs and MNPs

Transmission electron microscopy (TEM) images of the building block of these covalent complexes, namely, CD-3, AgNPs, and AuNPs, are provided in Figure 3a–c; size distribution histograms obtained from TEM images are given in Appendix A. CD-3 are spherical particles with an average size of 3.0 ± 1.1 nm; AgNPs are spherical particles with an average size of 13.4 ± 6.4 nm; and AuNPs have a polyhedral shape with an average size of 34.8 ± 7.4 nm. AgNPs also include a fraction of larger particles with an average size of 50.2 ± 15.8 nm (Appendix A). The data from the dynamic light scattering (DLS) measurements presented in Appendix A show that the hydrodynamic diameter of CD-3, AgNPs, and AuNPs is 5.8 ± 1.7, 10.1 ± 4.5, and 43 ± 17 nm, respectively. For AgNPs, a fraction of larger particles with an average size of 65 ± 22 nm was also observed in DLS (Appendix A), which agrees well with the TEM data. The hydrodynamic diameter of most of the studied species is larger than the TEM size due to the presence of ligands on the NPs’ surface [15]. In the case of AgNPs, the hydrodynamic radius was slightly smaller than that observed in the TEM images, which may be due to the non-spherical shape of that fraction. In order to study the surface composition of these samples, Fourier-transform infrared (FTIR) spectroscopy was employed. As shown in Figure 3d, CD-3 contained both primary amines (bands at 3450 and 3344 cm^−1^) and carboxylic groups (1710 cm^−1^) on their surface. The bands at 1580–1640 cm^−1^ can be attributed to either aromatic carbon network or -NH scissoring modes of -C-NH_2_ groups or amides [37], whereas the bands at 1370 and 1270 cm^−1^ can be attributed to C-O and C-N stretching modes. The FTIR spectra of AgNPs and AuNPs showed bands at 3200–3500 cm^−1^ and 1500–1600 cm^−1^, which can be attributed to N-H stretching and -NH_2_ scissoring modes, respectively, originating from cysteamine molecules at their surface.

Figure 3e shows the optical spectra of the building blocks of the covalent complexes. The absorption peak of CD-3 was centered at 450 nm, and the PL peak excited at 405 nm was centered at 570 nm. Two kinds of MNPs were chosen to probe two different enhancement regimes, with their plasmon peaks being in resonance with either absorption band (Figure 1b-i) or both absorption and PL bands of the CDs (Figure 1b-ii). For AgNPs, their plasmon resonance was located at 450 nm, which almost coincided with the CD-3 absorption peak and only had a small overlap with their PL band (Figure 3e); this situation corresponds to the first enhancement regime (Figure 1b-i). The broad plasmon resonance of AuNPs was located at 520 nm, overlapping with both the absorption and PL bands of CD-3, which corresponds to the second enhancement regime (Figure 1b-ii). The simulated plasmon resonance spectra of the two kinds of MNPs provided in Appendix A correspond well to the experimental ones (Figure 3e). The simulation of the electric field around the single MNPs (insets in Appendix A) indicate that the maximum enhancement at the MNP surface is expected to be around 16 and 6 times for AgNPs and AuNPs, respectively. For AgNPs, it is expected that the enhancement should occur through an increase in the excitation efficiency due to the effect (i) mentioned in the Introduction section, and for AuNPs due to the Purcell effect (effect (ii) in the Introduction section). As mentioned in the Introduction section, besides the enhancement of optical transitions by the electric field in the vicinity of MNP, FRET and plasmon-induced quenching may occur as well, which will be analyzed for the studied samples later on.

The formation of covalent complexes between the CD-3 and MNPs was accomplished by carbodiimide coupling reaction, which is a widely used method for the covalent bonding of various materials containing carboxyl and amino groups to form an amide bond [38]. *N*-(3-Dimethylaminopropyl)-*N*′-ethylcarbodiimide hydrochloride (EDC) was used to activate the carboxyl group on the surface of the CD-3 to create a crosslinking agent. *N*-Hydroxysuccinimide (NHS) was added to form a stable intermediate, which then reacted with the amine group at the MNP’s surface, as illustrated in Appendix A. Since CD-3 also have amines on their surface, there is a possibility of the formation of purely CD–CD complexes as well. This was examined by DLS, which showed the formation of aggregates with sizes of 14.5 ± 2.5 nm, corresponding to an agglomerate of 2–3 CDs (Appendix A). Thus, we assumed that CD–MNP complexes can be formed both from individual particles and from CD aggregates bonded to MNPs. To study the influence of the number of CDs conjugated to MNPs on the optical properties of the resulting complexes, two sets of samples (for AgNPs and AuNPs) with a weight ratio of CDs to MNPs (CD/MNP) varying from 0.1 to 10 were prepared.

TEM images of the CD–AgNP and CD–AuNP complexes shown in Appendix A and Figure 4a, respectively, demonstrate the presence of CDs in the close vicinity of MNPs, which manifested themselves as gray spots with interplanar distances of 0.21 and 0.24 nm, characteristic for carbon allotropes [39]. The DLS measurements showed that for CD–AgNP complexes with fewer attached CDs, the complexes tended to form agglomerates 8–10 times larger than AgNPs with an average size up to 165 ± 20 nm (Figure 4b). With an increase in the CD/AgNP weight ratio up to 10, the complexes with sizes of 18 ± 7 nm—only slightly larger than pristine AgNPs—were observed (Figure 4b), indicating that an interparticle distance reached 1–2 CDs in diameter. For the CD–AuNP complexes, a similar trend was observed, but with a smaller agglomerate formation: at the CD/AuNP weight ratio above 2, the size of the CD–AuNP complexes was almost the same as for the pristine AuNPs (Figure 4b). These observations indicated that stable conjugated CD–MNP complexes were preferentially formed at a rather high CD/MNP weight ratio with an interparticle distance of 1–2 CDs in diameter. From the DLS spectra of the CD–MNP complexes shown in Appendix A, there were signals from rather large agglomerates (>100 nm) whose size decreased upon an increase in the CD amount, which confirmed the stabilization of the complexes in solutions with an increase in the CD/MNP weight ratio.

FTIR spectra of CD–AgNP and CD–AuNP complexes are shown in Figure 4c; Appendix A shows them for samples produced with different CD/MNP weight ratios. The broad band at 3200–3500 cm^−1^ corresponded to -OH groups and H-bonding, while the bands at 1780 cm^−1^ and {1680, 1560} cm^−1^ indicated the presence of the C=O group and the amide group (C=O/C-N), respectively. Bands at 1515, 1500, and 1370 cm^−1^ in CD–AgNP and CD–AuNP disappeared, indicating the change in C-C and C-O bond configuration occurring during the EDC–NHS coupling reaction. Bands attributed to the -NH wagging mode at 710 and 670 cm^−1^ weakened due to the decrease in the amount of -NH groups. The band at 1280 cm^−1^ in pristine CD-3 attributed to the C-N group shifted to 1230 cm^−1^ in CD–MNP complexes, which can be explained considering that the amide group provides conjugation to a large particle, lowering its vibration frequency [40]. For CD-EDC/NHS conjugate and CD–MNP, similar sets of FTIR bands were observed, while the variation in the CD/MNP weight ratio only resulted in a slight difference of peak intensities with their positions preserved (Appendix A).

X-ray photoelectron spectroscopy (XPS) showed that the CD–MNPs complexes indeed included CDs with their C, O, and N peaks at 286, 533, and 401 eV, respectively (Figure 4d). The presence of MNPs manifested by XPS peaks at 373.8 and 367.7 eV for CD–AgNPs, and at 87.0 and 83.4 eV for CD–AuNP (Appendix A). High-resolution XPS spectra of C 1s, O 1s, and N 1s are shown in Appendix A. C 1s bands of all samples had three peaks at 285.0, 286.3, and 288.3 eV, corresponding to the C-C/C-H, C-OH/C-O-C, and C=O or O=C-N groups, respectively. An additional peak at 292.4 eV corresponding to the O-C=O bond also appeared for CD-EDC/NHS conjugates, CD–AgNPs, and CD–AuNP complexes (Appendix A). Compared to CD-EDC/NHS, the intensity of the O-C=O band decreased in the CD–MNP samples, pointing to the formation of an amide bond. The N 1s spectra revealed the presence of amine (399.3 eV) and amide (400.2 eV) bonds in all samples. In the XPS spectra of CD–MNP complexes, a peak at 401.6 eV attributed to an imide bond (C=O)-N-(C=O) emerged, confirming the formation of agglomerates comprising a few conjugated CDs. The O 1s spectra of both pristine CD-3 and both kinds of CD–MNP complexes had a peak at 532.3 eV, which was attributed to –O=C-N. In the CD–MNP complexes, the O 1s spectra also contained a peak at 531.2 eV, which can be attributed to the C-O bond in C-OH or C-O-C. We conclude that the XPS data coincides well with the FTIR data and confirm the formation of CD–MNP complexes via carbodiimide bonds.

### 3.3. Optical Properties of CD–MNP Complexes

In CD–MNP complexes, the increase in the CD/MNP weight ratio from 0.1 to 10 resulted in a linear growth of the intensity of the CD absorption peak at 420 nm (Appendix A), but in a nonlinear increase in the PL intensity (Appendix A). To evaluate the PL enhancement, the PL QY was estimated for CD–MNP complexes (Figure 5a,c) as a function of the CD/MNP weight ratio. For all of the CD–AgNP complexes, the PL QY was larger than that of pristine CD-3 (Figure 5a) with drop in PL lifetimes down to 1.5 ns compared to 2.5 ns for the pristine CDs (Figure 5b). A drop of PL QY at a CD/MNP weight ratio of 0.2 may indicate that most of CDs are bonded to the MNP surface directly, leading to the PL quenching. A further increase in the CD/MNP weight ratio (CD/AgNP > 5) resulted in a formation of stable colloidal CD–AgNP complexes with the largest (5.4-fold) PL enhancement (Figure 5a). The spectrum of electric field enhancement at the AgNP surface, calculated according to the Mie theory and presented in Appendix A, mostly overlapped with the CD absorption, which pointed out that the local amplification of the excitation light by MNP served as the main channel for the PL enhancement. The distribution of E-filed enhancement for AgNP shows a rapid decrease with the distance (Appendix A), and at the distance of the CD radius from the MNP surface, its average is equal to 5.6, which agrees well with the experiment. For CD–AuNP complexes, the electric field enhancement spectrum overlapped mostly with the CD emission, resulting in a different observed PL behavior upon increasing the CD/MNP weight ratio. At a low number of CDs per AuNP (CD/AuNP weight ratio < 0.2), the PL QY and PL lifetime dropped below the pristine CD level (Figure 5b,c). The increase in the CD/AuNP weight ratio up to 1 resulted in the PL lifetime decrease from 2.1 to 1.6 ns; this value then remained the same with a further increase in the CD amount. This observation can be attributed to the competing processes of PL enhancement by MNP and PL quenching via FRET from CD to MNP. The distance between the emitter and MNP with the highest PL quenching usually varies between 6 and 10 nm [25], corresponding well to the size of a single CD. Since that kind of quenching was absent in the CD–AgNP complexes, we consider the overlap between the AuNP plasmon resonance spectra and the CD PL band as the main reason for the observed effect. A further increase in the CD/AuNP weight ratio (CD/AuNP > 0.5) resulted in a gradual rise of PL QY together with a decrease in the PL lifetime. As a result, the overall PL enhancement reached its maximum (4.9-fold) at CD/AuNP = 5, matching well with the surface enhancement factor estimated at 532 nm (Appendix A). A further increase in CDs resulted in a decrease in the PL enhancement down to 3-fold, which can be caused by the limited distance of AuNP’s surface plasmon penetration into the surrounding medium, as illustrated in the inset of Appendix A and the E-field enhancement distributions at 400 and 532 nm (Appendix A). This observation pointed to the enhanced radiative recombination rate of CDs in the close vicinity of MNPs [41].

A schematic illustration of the electronic transitions of CDs in the close vicinity of plasmonic MNPs is given in a Jablonski diagram in Figure 6a. For the pristine CD-3, the relaxation of the lowest excited state (*S*_1_) can occur both through radiative (Γ) and nonradiative ($k_{nr}$) processes with $PL\;QY_{CD}=\Gamma /\left(\Gamma +k_{nr}\right)$. The estimated Γ and $k_{nr}$ values for CDs are equal to 4.0 × 10^6^ and 3.9 × 10^8^ s^−1^, respectively. If the CD’s intrinsic radiative decay rate remained unchanged in the CD–MNP complex, the change in PL QY can be attributed to the changes in $k_{nr}$, or to the plasmon-induced enhancement of Γ. The plasmon-induced radiative recombination rate, $\Gamma_{m}$, shortened the PL lifetime and affected the PL QY of the fluorophore near the metal surface according to $PL\;QY_{CD-MNP}=(\Gamma +\Gamma_{m})/\left(\Gamma +\Gamma_{m}+k_{nr}\right)$. This enhancement is related to the increase in the PL intensity of the CDs before the non-radiative energy dissipation, and it is more pronounced for fluorophores with low PL QYs [22]. Moreover, one should consider the PL quenching by reabsorption and/or near-field plasmonic dissipation prevailing the radiative relaxation of a fluorophore, and to account for this factor, another nonradiative constant, $k_{m}$, should be taken into account when estimating the PL QY of CD–MNP complexes: $PLQY_{CD-MNP}=\left(\Gamma +\Gamma_{m}\right)/\left(\Gamma +\Gamma_{m}+k_{nr}+k_{m}\right)$.

To estimate the influence of MNPs on the charge carriers’ relaxation in CDs in their complexes, normalized radiative and nonradiative rates were calculated as follows: $\Gamma_{m}^{*}=\left[(QY_{m}/\tau_{m})-\Gamma \right]/\Gamma$ and $k_{m}^{*}=\left[(1/\tau_{m})-\left(QY_{m}/\tau_{m}\right)-k_{nr}\right]/k_{nr}$. Figure 6b,c show calculated normalized $\Gamma_{m}^{*}$ and $k_{m}^{*}$ for CD–AgNP and CD–AuNP with different values of CD/MNP weight ratios. For both kinds of complexes, the nonradiative rate, $k_{m}$, was smaller than the intrinsic $k_{nr}$ (3.9 × 10^8^ s^−1^), with absolute values of 2.32 × 10^8^ s^−1^ and 1.42 × 10^8^ s^−1^ for the CD–AgNP and CD–AuNP complexes, respectively. The larger value of $k_{m}$ for CD–AgNP can be due to the stronger interaction and plasmon-induced quenching by a more intense AgNP plasmonic field than that observed for the AuNPs (see Appendix A). Thus, we can assume that the energy transfer that was expected to be the main nonradiative process of energy dissipation in the CD–AuNP complexes concedes the plasmon-induced quenching. For the CD–AgNP complex, the radiative rate $\Gamma_{m}$ was larger than the intrinsic one (4.0 × 10^6^ s^−1^) for all values of the CD/MNP weight ratio (Figure 5b), reaching 29.5 × 10^6^ s^−1^ for CD/AgNP = 10. For the CD–AuNPs, $\Gamma_{m}$ became larger than the intrinsic radiative rate (4.0 × 10^6^ s^−1^) when the CD/AuNP was >0.5 with a maximal value of 23.8 × 10^6^ s^−1^. We note that for both CD–AgNP and CD–AuNP complexes, the radiative rate $\Gamma_{m}$ increased with CD/MNP weight ratio almost linearly (Figure 6b,c), which may appear to be counterintuitive. However, considering the assumed buildup of the obtained CD–MNP complexes, whose diameter roughly corresponded to the sum of the average size of MNPs plus the thickness of one layer of CDs (an interparticle distance of 3–6 nm), as observed for CD/MNP > 0.5 (as shown in Figure 4b), larger PL enhancement and $\Gamma_{m}$ values were obtained. We can conclude that the actual morphology of the CD–MNP complex played an important role in the PL enhancement, whereas an increased CD/MNP weight ratio contributed to a better stabilization of those complexes.

## 4. Conclusions

We showed that the interactions of CDs and MNPs and their optical responses in CD/MNP complexes depend on several parameters: (i) the type of formed bonds between CDs and MNPs; (ii) the architecture of CD–MNP complexes, including their size and the number of CDs per MNP; and (iii) the degree of overlapping of absorption and emission of CDs with the plasmon resonance of the MNP. Concerning point (i), the interactions of CDs with MNPs in covalently bonded complexes resulted in a larger PL enhancement (up to 540%) compared to complexes formed by the dynamic interaction of CDs with MNPs by Coulomb attraction, where only a very minor 10% enhancement was achieved. Concerning point (ii), in the stable covalent CD–MNP complexes with anticipated architecture corresponding to MNPs covered by several layers of CDs, considerable PL enhancement of up to 5.4- and 4.9-fold for CD–AgNP and CD–AuNP complexes has been achieved. Concerning point (iii), regardless of the optical enhancement regime, e.g., absorption-only or the absorption and emission of CDs overlapping with the plasmon resonance of MNPs, non-radiative losses were caused by the plasmon-induced PL quenching rather than FRET from CDs to MNP. At the same time, the emission enhancement was of a plasmon-induced nature, and it was determined by the size/material of the MNP and the distances between the particles’ surfaces in the CD–MNP complexes. Since the optical transitions of CDs were influenced by covalently bonded MNPs, which resulted in increased PL QYs and decreased PL lifetimes, the CD–MNP complexes studied in this work can be used in various applications in biomedical and photonic fields.

## Figures and Tables

**Figure 1 nanomaterials-13-00223-f001:**
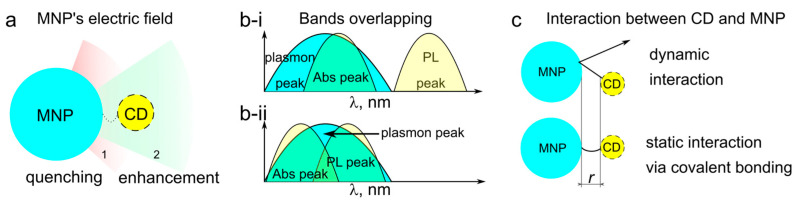
CD and MNP interactions. (**a**) Schematic representation of the influence of the distance between CD and MNP on quenching/enhancement of the CD’s emission. (**b**) Excitation regimes: (i) only the absorption (Abs) band is in resonance with the plasmon peak; (ii) both absorption and PL bands are in resonance with plasmon peak of MNP. (**c**) Interactions of CD and MNP: dynamic by attractive and repulsive forces in colloidal solution; static via covalent bonding of CD to the surface of the MNP.

**Figure 2 nanomaterials-13-00223-f002:**
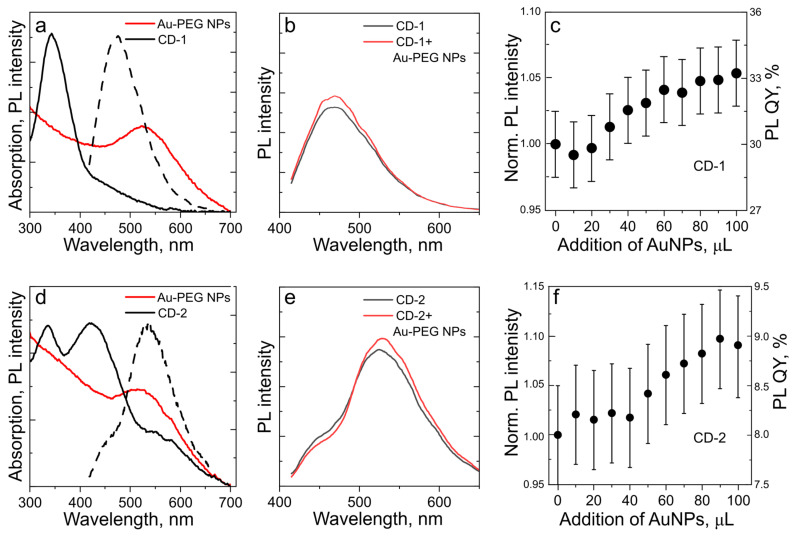
Dynamic interaction of Au-PEG NPs with CD-1 (**a**–**c**) and CD-2 (**d**–**f**): (**a**,**d**) overlapping of absorption (black line) and PL spectra excited at 405 nm (dashed black line) of CDs with absorption of Au-PEG NPs (red line); (**b**,**e**) PL spectra excited at 405 nm of CDs without (black line) and with Au-PEG NPs (red line); (**c**,**f**) change of the integrated PL intensity of CDs normalized to the intensity of pristine CDs (PL enhancement) and PL QY upon addition of increasing amount of Au-PEG NPs.

**Figure 3 nanomaterials-13-00223-f003:**
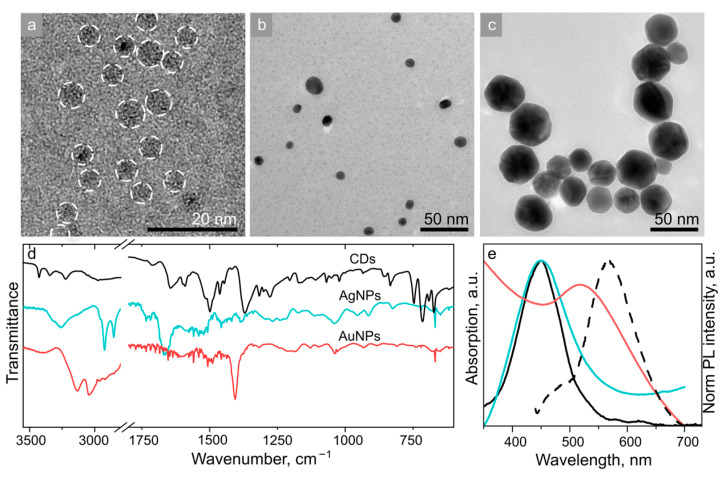
TEM images of (**a**) CD-3, (**b**) AgNPs, and (**c**) AuNPs. (**d**) FTIR and (**e**) absorption spectra of CD-3 (black), AgNPs (cyan), and AuNPs (red). PL spectrum of CD-3 excited at 405 nm is also shown on the frame (**e**) by a black dashed line.

**Figure 4 nanomaterials-13-00223-f004:**
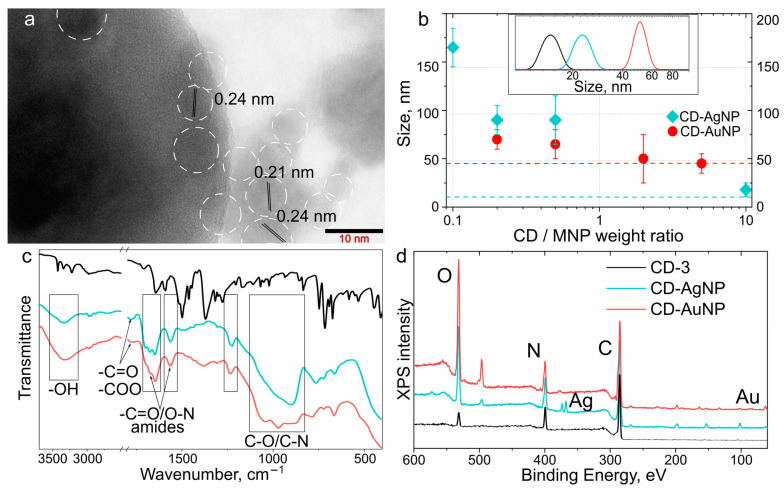
(**a**) TEM image of the CD–AuNP complex, with an indication of the interplanar distances attributed to CDs. CDs are highlighted by dashed circles. (**b**) Hydrodynamic diameters of the CD–AgNPs complexes (cyan diamonds) and CD–AuNP complexes (red circles) estimated from DLS measurements, as a function of the CD/MNP weight ratio. Dashed cyan and red lines indicate the values of hydrodynamic diameters of pristine AgNPs and AuNPs, respectively. Inset exemplifies typical size distribution curves determined by DLS for the CD-EDC/NHS conjugate (gray line), CD–AgNP complex (cyan line), and CD–AuNP complex (red line). (**c**) FTIR and (**d**) XPS full survey spectra of CD-3 (black line), CD–AgNP complex (cyan line), and CD–AuNP complex (red line).

**Figure 5 nanomaterials-13-00223-f005:**
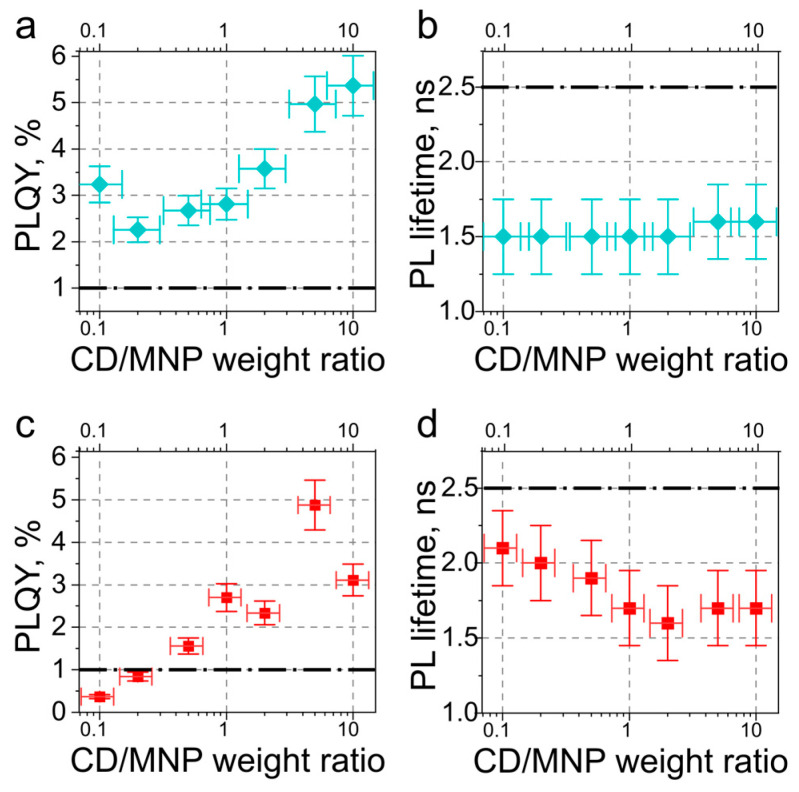
Optical properties of (**a**,**b**) CD–AgNP complexes and (**c**,**d**) CD–AuNP complexes. PL QY (**a**,**c**) and average PL lifetime (**b**,**d**) versus CD/MNP weight ratios. Dash-dot lines in (**a**,**c**) and (**b**,**d**) provide the values of PL QY and PL lifetime for the pristine CD-3, respectively.

**Figure 6 nanomaterials-13-00223-f006:**
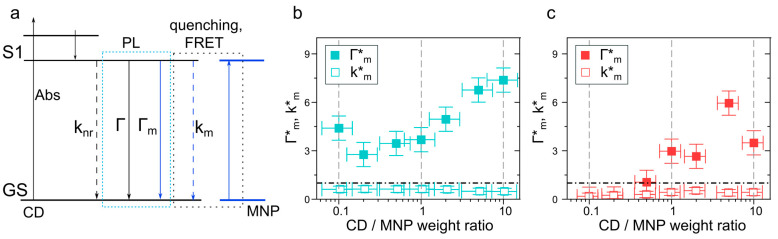
(**a**) Jablonski diagram illustrating transitions occurring in the CD–MNP complex. After absorption (Abs) of the incident light by CDs, it rapidly (10^−12^–10^−10^ s) relaxes to the lowest excited state, S1, from which several pathways of radiative (PL) and nonradiative relaxation to the ground state (GS) can occur. Γ and $k_{nr}$ are rates of radiative and nonradiative relaxation of the pristine CDs; $\Gamma_{m}$ and $k_{m}$ are rates of radiative and nonradiative relaxation of CDs in the presence of MNP. Experimentally determined radiative ($\Gamma_{m}^{*}$, closed squares) and nonradiative ($k_{m}^{*}$, open squares) rates for (**b**) CD–AgNP and (**c**) CD–AuNP complexes normalized by values of $\Gamma_{m}$ and $k_{m}$ for the pristine CD-3 (shown by dash-dot lines) versus CD/MNP weight ratio.

## Data Availability

Not applicable.

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
