# Peer review of "Carbon Dot Emission Enhancement in Covalent Complexes with Plasmonic Metal Nanoparticles"

_nanomaterials, 2023, doi:10.3390/nano13020223_

Round 1

Reviewer 1 Report

The manuscript by Arefina et al. presents a study of the optical properties of the complexes between carbon quantum dots and plasmonic nanoparticles. The authors use two types of chemical binding (an electrostatic interaction and a covalent binding) and variety of optical and physicochemical characterization techniques to study the morphology of the complexes and the pathways of the photoexcitation energy migration in the complexes. The authors demonstrate significant (5.4 and 4.9-fold) enhancement of the PL of the carbon dots in the covalently-bound complexes with silver and gold nanoparticles, respectively.

I very much liked the study as well as its presentation by the authors. Fine-tuning of the optical transitions of the carbon quantum dots by placing them close to the plasmonic nanoparticles as shown by the authors provides important new possibilities for the plasmonic biosensing, and are thus of considerable, broad interest for nanotechnological and optical applications.

I have a couple of minor comments/suggestions:

1.)        Figure 3a and Figure S6 show TEM images of the complexes of silver /gold nanoparticles and the carbon dots. However, the morphology and stoichiometry of the complexes can not be distinguished from these images, mostly because of the low contrast of the carbon dots in the TEM images. It may be nice to highlight the carbon dots in the images. In addition, the paper will benefit from the larger-scale TEM images showing several complexes in one image. This could directly visualize the distribution of the complexes’ morphology.

2.) Section “Conclusions” page 10 line 390: “At the same time, the emission enhancement was of the plasmon-induced nature, and it was determined by the size/material of MNP and interparticle distances in the CD-MNP complexes. “ It might be worth evaluating the interparticle distances in the complexes to support this conclusion.

Reviewer 2 Report

In this manuscript, stable colloidal complexes based on carbon dots and plasmonic nanoparticles are developed through covalent bond connection. When the coupling of carbon dots with silver and gold nanoparticles are in the most optimized architecture, 5.4 and 4.9 times of photoluminescence enhancement can be achieved. This research study expands the toolkit of functional materials based on carbon dots for applications. A revision addressing the following points is needed.

1.        The existing research reports show that the radiation rate under the effect of the Percell effect is mainly determined by the excitation power, wavelength, local density of states, etc. In order to prove the existence of the Percell effect, the manuscript should provide PL intensity dependent (excitation power dependent) PL lifetime measurement.

2.        In addition to FRET mentioned in the manuscript, the existing report "Leaving Forster Resolution Energy Transfer Behind: Nanometal Surface Energy Transfer Predicts the Size Enhanced Energy Coupling between a Metal Nanoparticle and an Emerging Dipole" indicates that NSET dominated energy transfer is widespread in small metal particles. Please discuss this.

3.        In Fig. 1 (a), how are the positions of enhancement and quenching between CD and MNP determined? Please explain.

4.        In Figure 4, curves corresponding to different CD/MNP weight ratios are not clearly marked.

Reviewer 3 Report

Authors report protocols for and the effects of carbon dots (CDs) and Au/Ag nanoparticles complex formation on the optical properties of these new systems. They also state that they "...demonstrate how interactions between carbon dots and metal nanoparticles in the formed complexes, and thus their optical responses depend on the type of the bonds between particles, the architecture of complexes, and the degree of overlapping of absorption and emission of carbon dots with the plasmon resonance of metals."

I found that the manuscript is, in general, well written, the motivation for the work is clearly explained and the approach is well-outlined and sufficient details are given. Experimental work is described in sufficient details to reproduce the results and good details are provided for synthesis and characterisation. The key questions and findings are clearly-argued in the Results section and supported by an adequate discussion. I found that conclusions are adequately supported by the corresponding experimental evidence.  

As a consequence, this work is certainly of interest for those working in the area of CD research as well as for the fields that benefit from development of novel fluorphores.

At the same time, there are a number of areas where I think the manuscript will require corrections/improvement before publication. I outline these areas below:

1. There are a few generic (to the point of being meaningless) statements  in the abstract (the latter must describe key specific findings). For example, in the sentence "In particular, development of stable colloidal complexes based on carbon dots and plasmonic nanoparticles allows to achieve synergistic effects and expand their physical and chemical properties." it is not clear what "synergistic effects" authors mean specifically - bringing more any two (or more) systems together will always result in synergistic effects. The same is true for expansion of physical properties - that statement means nothing without specific examples. I'll recommend removing the sentence.  

2. I would suggest changing "... such as excitation-dependent (or independent) emission with a high photoluminescence quantum yield (PL QY)." to "... such as excitation-dependent (or independent) light emission with a high photoluminescence quantum yield (PL QY)". 

3. in the "Experimental Setup" section is is not clear what the i-th component is in the expression for PL time decay: i-th component of what? This needs to be clarified.

4. In the lines140-142 the sentence construction is slightly ambiguous and awkward. I would recommend changing these to "... with absorption at 345 nm and PL bands at 475 nm (Figure S1a), and CD-2 synthesized according to Ref.[30] whose absorption at 425 nm and PL bands at 535 nm (Figure S1d) have been chosen. "

5. in lines 156-157 change "... oppositely charged surface of these two kinds of nanoparticles (CDs and MNPs)..." to  "oppositely charged surface of these nanoparticles (CDs and MNPs)...".

6. I do not agree that "TEM images of CD-AgNP and CD-AuNP complexes shown in Figure S6 and Figure 3a, respectively, demonstrate the presence of CDs in the close vicinity of MNPs...". At least not without some convincing arguments that it is the case. All I can see in Figs S6 and 3a are some nanoscale objects, but it is impossible to conclude convincingly what they are simply on the evidence of the pictures. Furthermore, the quality of Fig. 3a is such that I do not see any interplanar distances that authors suggest they observe. Better evidence for such distances is required e.g. by providing better quality picture with clear fringes and/or carrying Fourier transform of the images in the area of interest. I general, I find that TEM data provided by the authors are not sufficient to demonstrate CD-MNP complex formation. However, other data (e.g. FTIR, XPS) do seem to support the argument for complex formation. Unless better TEM evidence is provided along the lines I described above, I suggest removing argument that TEM provides evidence for complex formation.

7. In lines 228-229 replace "... complexes with less number of attached CDs ..." with  "... complexes with fewer attached CDs ...". 

8. Authors rightfully chose to investigate the dynamic interaction (DI) of CD-MNPs before moving on investigation of the covalent complexes (CC). Such comparative analysis provides crucial insights into the effects of CD-MNP bonding on their optical properties. It is then not clear to me why optical data for DI (Fig. S1) are given in Supporting Information (SI), rather than in the main article (like the CC optical data, Fig. 4). Furthermore, I would like to see an explanation why in the case of DI authors give the PL enhancement in terms of PL intensity changes, while in case of CC a more appropriate PL QY is given - one would expect a consistent approach across these two cases investigated by the authors. Thus, I would like to see PL QY and PL lifetime data for DI samples. This is important as the key deferences between these data would point to the origins of the observed changes in the optical properties. For example, comparison of PL QY data would provide a clear and convincing evidence of the effects of covalent bonding on the light emission, while right now trends look very similar (compare e.g. Fig. S1c and Fig. 4c), suggesting that the nature of the changes may also be similar. I would also suggest moving panels a, b, e and f in Fig.4 into SI as these play largely supporting role, while PLY QY and PL lifetime data provide critical information in the context of the discussion.

I believe these suggestions amount for a minor revision and that is what I recommend. 
